# Dry Anaerobic Digestion Technologies for Agricultural Straw and Acceptability in China

**Yanran Fu, Tao Luo \*, Zili Mei, Jiang Li, Kun Qiu and Yihong Ge**

Biogas Engineering Research Center, Biogas Institute of Ministry of Agriculture (BIOMA), Chengdu 610041, China; 18302871287@163.com (Y.F.); 13880233242@163.com (Z.M.); lijiangdream@hotmail.com (J.L.); qiukun@caas.cn (K.Q.); geyihong@caas.cn (Y.G.)

\* Correspondence: luotao@caas.cn, Tel.: +86-28-85226493, Fax: +86-28-85287726

**Abstract:** Dry anaerobic digestion technology (DADT) is considered a highly feasible way to treat agricultural straw waste; however, most practical operations are always in low efficiency, due to the poor fluidity behavior and complex lignocellulosic structure of straw, which is not easily decomposed by anaerobic bacteria. Hence, it is necessary to further investigate the operation boundary, in order to increase biogas production efficiency for effective applications. In this paper, typical DADTs are reviewed and their suitability for application in China is analyzed. The advantages and disadvantages of different anaerobic digestion processes are evaluated considering pretreatment, organic loading rate, anaerobic digestion temperature, and homogenization of the feedstock and inoculate. The suitability of the DADTs is evaluated considering the accessibility of straw resources and the convenience of biogas use. It is concluded that batch anaerobic digestion processes would be more suitable for the development of southern China due to the prevalence of small-scale agriculture, while continuous anaerobic digestion would be preferable in the north where large-scale agriculture is common. However, the DADTs discussed here need to broad application in China.

**Keywords:** biogas; operation; review; straw collection; biogas utilization

## 1. Introduction

In China, approximately 0.9 billion tons of agricultural straw was generated in 2015, which is considered as one of the most abundant sources for producing renewable energy [1]. Unfortunately, a large proportion of the produced straw is disposed of by burning the fields directly or tilling back into the soil, which cause serious environmental problems, such as air pollution and soil degradation [2]. According to the statistics of Chinese government, 20% of the straw could be utilized for anaerobic digestion, and $5 \times 10^{10}$ m$^3$ biogas could be produced, due to technical feasibility and feedstock accessibility [3]. Hence, there is the opportunity to change from a situation of poor resource management to efficient use of straw as a bio-resource, a suitable technology make the transition for these residues [4–6]. As agricultural systems vary in different regions, that results in an inconsistent geographical distribution of straw, the treatment processes should also be suitable for sustainable development.

Biogas seems to be a more effective way, compared with other renewable ways, such as bioethanol, to treat agricultural straw and produce renewable energy simultaneously [7]. Meanwhile, it could be feasible at many scales from a household to large biogas plants, its energy yield could be much higher for energy crops since 2005, the Chinese government has initiated demonstration projects in 11 provinces [8]. During the period of 2011–2015, the development of straw biogas plants experienced a boom, and the digestion volume greatly increased, up to 3000–5000 m$^3$ for a single unit [9,10]. Over the

period of 2016–2020, the Chinese government will continue to develop biogas technology and plans to build 20,000 biogas projects to treat a larger fraction of the agricultural straw produced locally [11].

Anaerobic digestion technology is divided into wet and dry anaerobic digestion by the total solids (TS) content, where a value ≤ 15% is defined as wet anaerobic digestion, while >15% is considered dry anaerobic digestion [12]. Wet anaerobic digestion has lower operating efficiency and higher costs, due to the higher moisture of the straw (TS of dry corn stover > 80%; TS of silage corn stover > 30%). Although, wet digestion is still the first choice for most biogas plants due to the low-tech equipment [13,14]. Nevertheless, it has problems related to the large quantities of extra water required, large digestion volume and distribution low TS content digestate [15–17]. In addition, the formation of floating layers is an obstacle for achieving efficient mixing and biogas release [18]. All of these seriously limit the volumetric biogas production and decrease the energy conversion efficiency of straw.

In order to overcome the disadvantages discussed above, technologies are considered preferable for treating straw, as these processes have a lower water input, smaller digestion volume, and higher TS content of the digestion waste compared to wet processes [19]. The dry anaerobic digestion process does not present problems of foam, sedimentation, surface crust, and does not require the reduction of size, or removal of inert materials and plastics [20]. Furthermore, DADTs could reduce energy input for biogas production in the winter by 10% to 15% [7]. Therefore, it is meaningful to review the present DADTs being developed and used in China and abroad, in order to identify their suitability for widespread use in China.

The purpose of this paper could promote the acceptability of DADTs in China, which provide a maneuverable and economical method for the utilization of agricultural waste. The remaining sections of this article are arranged as follows: Section 2 describes the evaluation indicators, which would affect the stability and efficiency of the dry anaerobic digestion process, and introduces the various types of DADTs, including operating principle and technical features; Section 3 analyzes DADT applications in China according to the amount of resources, geological conditions and biogas utilization; and Section 4 presents the main conclusions and recommendations.

## 2. Technical Evaluation of Dry Anaerobic Digestion Processes

Dry anaerobic digestion includes batch and continuous processes, and the applied option is determined by the actual situation. The main advantages of batch processes include low operation and maintenance requirements, while continuous processes operate in a relatively steady state and have quite constant biogas production rates [7]. In order to be considered for technical applications, DADTs must meet the optimal straw digestion parameters, which is the key point for operation efficiency and biogas production.

### 2.1. Evaluation Indicators

Straw is a kind of lignocellulosic material consisting of three major components: Cellulose, hemicellulose, and lignin, where most of the cellulose is combined with lignin and hemicellulose [21]. As the main limiting factors for achieving high biogas production rates are related to the rheology and hydrolysis process, the resistant lignocellulosic structure does not digest well [22]. Therefore, several processes are generally undertaken to improve digestion efficiency, including: (1) Pretreatment of straw to increase the hydrolysis rate [23]; (2) selection of an appropriate organic loading rate (OLR) to maximize the net yield of energy production and restrict inhibition [24]; (3) controlling the digestion temperature, which is essential to digest, especially for the biodegradation rate optimization [25, 26]; and (4) efficient mixing to enhance contact between the microorganisms and substrates [26]. These processes are discussed in detail in the following sections.

2.1.1. The Degree of Hydrolysis

Pretreatment mainly includes physical (i.e., size reduction, thermal), chemical (i.e., acid and alkali addition), and biological (microbial degradation) methods, which increase the degradability

and accelerate the degree of hydrolysis [27]. Lizasoain et al. studied the effect of steam explosion on the chemical composition and biomethane potential of corn stover using temperatures of 140–220 °C and pretreatment times of 2–5 min, where a pretreatment at 160 °C for 2 min improved the methane yield by 22% [28]. Mustafa et al. used rice straw, pretreated by physical (milling to ≤ 2 mm) and biological (incubation with *Pleurotus ostreatus* fungus) methods, to improve the degradability and biogas production, achieving a 165% increase of methane yield compared to untreated samples [24].

Although these pretreatments could increase biogas yield to some extent, there are also unavoidable disadvantages. Physical pretreatment always requires high-energy inputs and high equipment investment costs [25]. Biological pretreatment requires specially trained employees to perform the process [29]. It seems to be very difficult to reuse the digestion residue as fertilizer after chemical pretreatment. Based on these limitations, milling is thought to be the most viable method as it is a simple process [27]. In China, after a combine harvester is used to reap the straw, milling equipment could be used to crush the straw down to the length of 5 cm generally [30]. Therefore, it is essential to investigate DADTs for digesting such particle straw.

### 2.1.2. Organic Loading Rate (OLR)

The OLR has a significant influence on the stability of the anaerobic digestion of straw [31]. High OLRs result in excellent growth of *fibrobacteres* bacteria due to the abundant feedstock and hence a high methane production rate, however, the concentration of volatile fatty acids (VFA), isobutyrate, isovalerate, and other toxic substances can be increased simultaneously. In the case of low OLRs, the metabolism and biogas production rate of the microbes is limited due to inadequate nutrition [32]. Zealand et al. investigated the effect of OLR on anaerobic digestion and showed that an OLR of 1 g VS/(L·d) achieved a volumetric biogas production rate of 300 ml/(L·d) with 50 % $CH_4$ content; while in the case of 2 g VS/(L·d), 300 ml/(L·d) production with 52% $CH_4$ was observed. Hence, increasing the OLR does not inevitably increase the biogas yield in the same proportion [33]. Therefore, the OLR used in a particular DADTs process needs to be carefully optimized in order to achieve high digestion efficiency.

### 2.1.3. Temperature

Temperature is a key factor that needs to be considered in the anaerobic digestion process, not only for maximizing the efficiency and energy production, but also for reducing energy inputs (and hence, operating costs) [34]. Both mesophilic (around 35–40 °C) and thermophilic (55–60 °C) conditions have been used for biogas digestion, and the operational performances under these conditions have been widely reported [35]. Labatut et al. indicated that thermophilic processes are prone to inhibition and instability, as pH increases and ammonia production becomes unstable, which is generally toxic to the methanogens [36]. The mesophilic process is more robust and less sensitive to changes due to the higher diversity and richness of bacteria coexisting inside the reactor [37]. In addition, lower energy and maintenance costs are advantages, promoting the mesophilic condition to be accepted by most biogas plants.

### 2.1.4. Mass Transfer

Straw is characterized by its low bulk density, high-water holding capacity, and poor fluidity, resulting in poor heat and mass transfer, which results in detrimental non-homogeneity of the mixture that degrades digestion performance [29,38]. Mixing is deemed as very important for achieving a good distribution of the substrates, microorganisms, and enzymes in the digester, in order to optimize anaerobic digestion [39]. Ivoachu et al. confirmed that recycling percolate in dry anaerobic digestion is a viable method for digesting straw more rapidly and effectively, due to the higher nutrient and microbial concentration in the recycled percolate [38]. Computational fluid dynamics (CFD) is commonly used to calculate the flow distribution in digestion systems, in order to determine the suitable stirring parameters. For example, Shen et al. found that the biogas production rate could be increased using

triple impellers with pitched blades, which achieved effective mixing at a stirring rate of 80 rpm; this indicates that proper agitation is a cost-effective way to enhance anaerobic digestion [40].

### 2.2. Batch Anaerobic Digestion

#### 2.2.1. BEKON System

Figure 1 shows a schematic diagram of the typical BEKON batch digestion system [41]. This technology is also referred to as the 'garage-type' percolation batch reactor and is mainly used for agricultural straw and yard waste. In this system, leachate is returned to the digester through a pump, then it is sprayed on a material surface. Optimizing the leachate backflow conditions, such as interval time and amount, is key for managing the process as it optimize microorganism metabolism and increase biogas yield through mass transfer process. Advantages of this process include: (1) Low maintenance requirements and little systemic energy loss [42]; and (2) high TS content fermentation residues (bio-fertilizer) can be obtained directly, which avoids solid–liquid separation processes and minimizes the treatment of waste liquid [43,44].

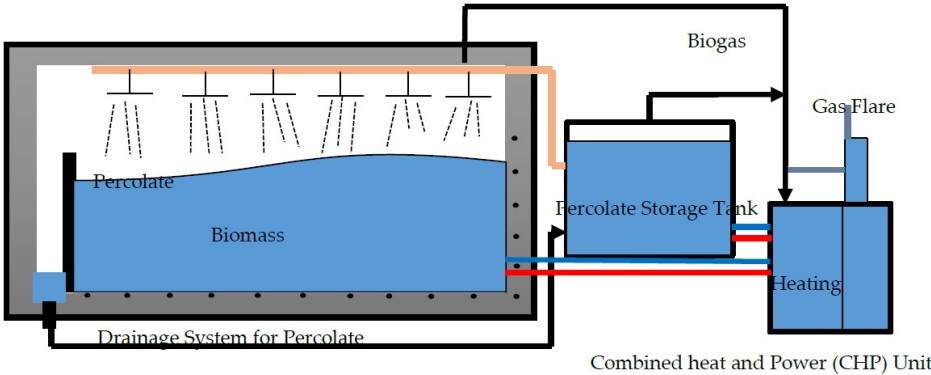

**Figure 1.** Schematic diagram of the BEKON batch system design with percolate.

#### 2.2.2. GICON System

Figure 2 shows a schematic diagram of the batch process implemented by GICON Holding GmbH, which is operated without inoculation, but by recirculation of the percolate. The percolate is stored in a tank, and supplied to an external digester, where soluble material is further degraded and converted into biogas; this is the main difference compared to the BEKON system. The GICON system could be classified as a two-phase digestion system, which divides the biogas production into acetogenesis and methanogenesis stages in order to ensure stable operation and better performance; the risk of acidification is effectively avoided, and accurate process control is achieved [43,45]. The advantages of this process include: (1) Straw with a large particle size or no straw in the acetogenesis system [46]; (2) a simple and highly effective methanogenesis system due to reutilization of the percolate [47]; and (3) a higher methane yield than the BEKON system due to the specific activity of the methanogens and reduced the yield of biogas slurry. The performance of dry fermentation biogas plants is carefully monitored in Harbin, where both BEKON and GICON technology have been adopted, indicating that such technology is appropriate for application in China.

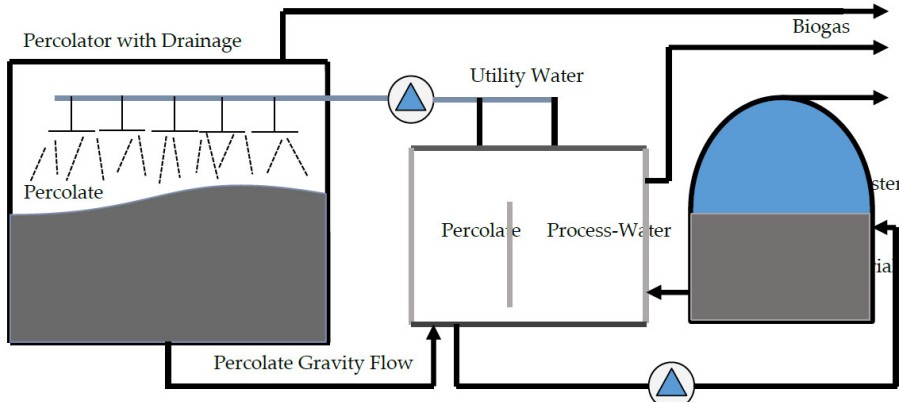

**Figure 2.** Schematic diagram of the GICON batch process with percolate.

### 2.2.3. BIOFerm System

Figure 3 shows a schematic diagram of the BIOFerm system [45]. In this case, the TS content of the feedstock mound is about 25% to 35% and, the percolate recirculation process is designed to recover residual heat from combined heat and power. The advantages of this system are as follows: (1) Low pretreatment costs due to use of large feedstock particles [46]; (2) effective use of heat to maintain fermentation temperature and reduce energy input; and (3) high-TS-content organic fertilizer is obtained as a valuable byproduct of the anaerobic digestion [47]. Kasakova et al. [45] reported a BIOFerm system treating corn silage, grass silage, and beef cattle manure, with a TS content of 20%, resulting in a methane yield of 0.111 m$^3$/kg VS.

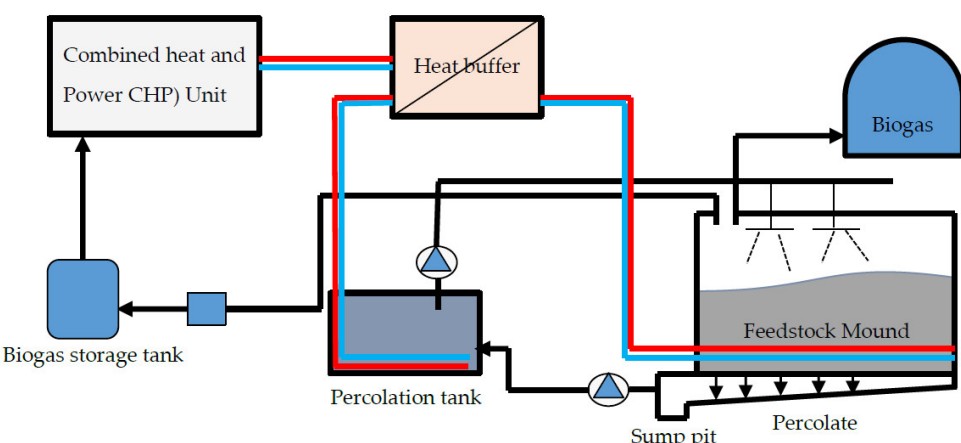

**Figure 3.** Schematic diagram of a BIOFerm batch processing system with percolate.

### 2.2.4. Chinese Batch Reactors

The predominant DADT used in China is a membrane-covered trough system (MCT), which is equipped with garage-style flexible roof membranes (GFRM), as shown in Figure 4 [48,49]. This system can operate with 23% to 40% TS. During practical operation, the required amount of biogas is produced by adding or reducing the number of MCT bioreactors. Another common garage-style DADT used in China is shown in Figure 5, which is combined with a leachate recirculation system. The volumetric biogas production of these systems is limited as they lack an effective agitator to achieve sufficient mixing of the substrates and microorganisms. Although the products have undergone considerable development, in most cases the technologies are in the prototype stage and have not yet been tested on an industrial scale.

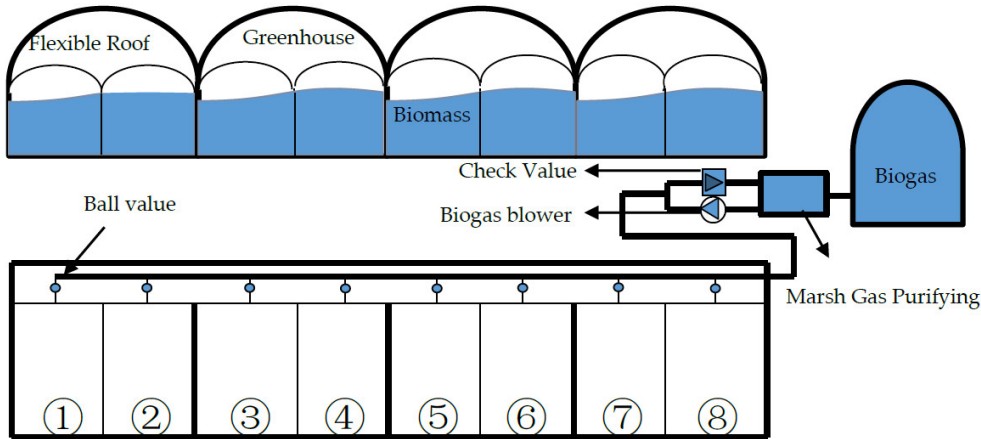

**Figure 4.** Schematic diagram of a film-covered trough biogas dry anaerobic fermentation system.

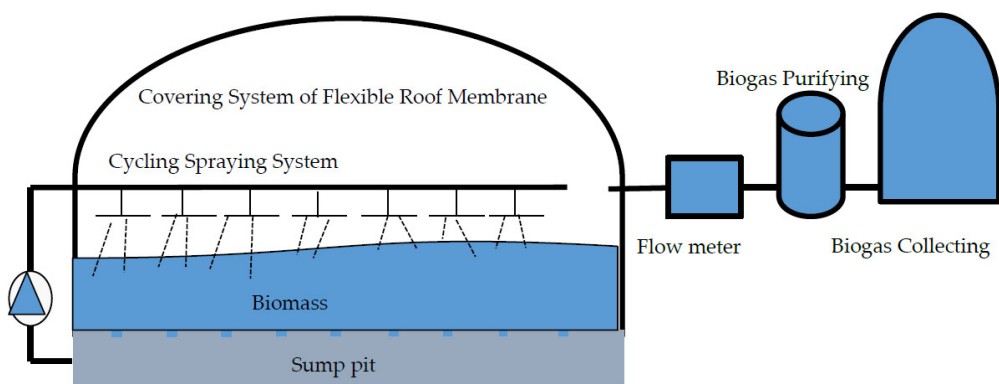

**Figure 5.** Schematic diagram of dry fermentation equipment based on the garage-style system with a flexible roof membrane and a percolate recirculation system

### 2.2.5. Comparison in Technology

Table 1 summarizes the operating performances of the main batch anaerobic digestion processes. From Table 1, we present the following conclusions: (1) Pretreatment is unnecessary condition for batch digestion technology; (2) GICON fermentation system can operate with high TS contents, as it can avoid accumulation of VFAs effectively by the two-phase system which separates the stages of acetogenesis and methanogenesis into two parts; however, it is more complex than the other systems; (3) recirculation and spraying of the percolate can improve the mass transfer and biogas production rate; (4) Chinese batch DADTs still need a long road to undergo and have shown relatively low biogas production efficiencies; and (5) discontinuous biogas production is a critical barrier for stable biogas supply.

**Table 1.** Comparison of the operating performances of various biogas batch technologies.

| Reactor | Country | Capacity (t/year) | Substrate | TS (%) | OLR | T (°C) | HRT (d) | Methane Yield ($Nm^3_{CH4} \cdot kg_{vs\ removed}{}^{-1}$) | Methane Average (%) | Ref. |
|---|---|---|---|---|---|---|---|---|---|---|
| BEKON | GER | 7500–40,000 | AW | N/A | N/A | 37–55 | 28–35 | 0.17–0.37 | 52–62 | [39] |
| GICON | GER | 30,000–40,000 | AW, OFMSW | 36 | N/A | 37 | 35 | N/A | 53 | [50] |
| BIOFerm | GER | 8000 | AW, OFMSW | 25 | N/A | 37 | 28 | 0.21-0.35 | N/A | [39] |
| MCT | PRC | N/A | AW | 10–20 | N/A | 35–37 | N/A | N/A | 55–60 | [51] |
| GFRM | PRC | N/A | AW | ≥8 | N/A | 37 | N/A | N/A | N/A | [52] |

N/A: Not available, AW: Agricultural waste, OFMSW: Organic fraction of municipal solid waste, T: Process temperature, HRT: Hydraulic retention time.

### 2.3. Continuous Digestion

#### 2.3.1. Dranco System

Schematic diagram of the Dranco process is shown as Figure 6, as designed by Organic Waste Systems (OWS) of Belgium. This system uses a vertical silo with a conical bottom as reactor, with a conical bottom with an auger used to collect percolate, and a mixing unit is used to mix the raw feedstock and anaerobic microorganisms prior to biogas digestion, as the digester does not have an internal mixing mechanism [49]. Advantages of this process include: (1) Mixing is finished outside the digestion system, the residence time of materials in the reactor is shortened [53]; (2) a high degree of percolate recirculation effectively recycles anaerobic bacteria and waste heat; and (3) a TS content of 30% to 40% is maintained over long-term operation [53].

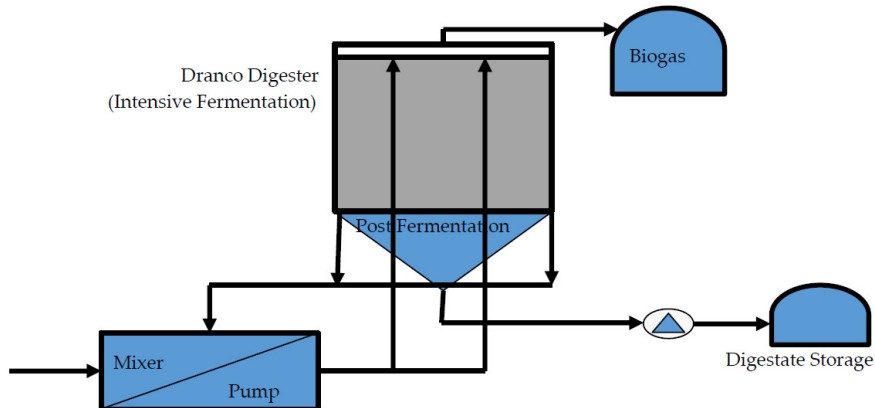

**Figure 6.** Schematic diagram of the continuous Dranco process.

#### 2.3.2. Linde-KCA System

The Linde-KCA system is operated as a plug flow reactor incorporating aerobic and anaerobic digestion in separate tanks, as shown in Figure 7 [54]. The TS content of the feedstock can be between 15% and 40% [50]. Several axle mixers agitate the slurry and increase homogenization. The characteristics of this process include: (1) Impellers within the digesters result in good substrate mixing, enhance access of the microorganisms to the substrates, and hence efficient substrate conversions; (2) high OLR can be realized due to the effective mixing; and (3) there is an additional operating cost related to the aerobic pretreatment. Patinvoh et al. [55] investigated the performance using untreated manure embedded with straw at 22% TS for 230 days. They found that an OLR of 4.2 g VS/(L·d) resulted in the most stable operation, with methane yields up to 0.163 L $CH_4$/gVS added [55].

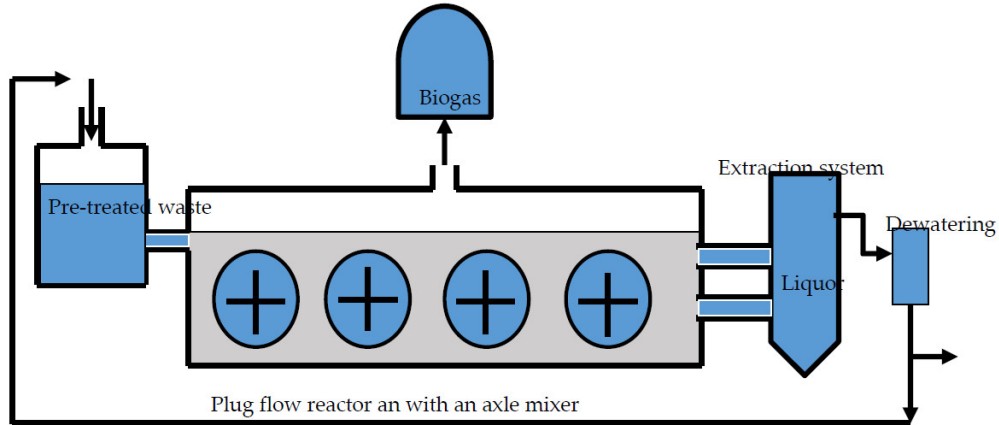

**Figure 7.** Schematic diagram of the Linde-KCA continuous system.

### 2.3.3. Valorga System

The Valorga process is shown as Figure 8, which consists of a cylindrical vertical digester with a horizontal plug flow system. The reactors contain a vertical median inner-wall on approximately two thirds of their diameter, which achieve "plug-like" flow in the reactor [42]. The inlet for feeding and outlet for discharging the substrate are located on the middle and lower segments of the main wall, respectively. The central wall functions as a baffle that extends two thirds of the diameter through the center of the tank, where the material is forced to flow around the baffle from the inlet to reach the outlet on the opposite side, creating a plug flow in the reactor [56]. The main characteristics of this process are: (1) The central wall enhances circular flow of the substrate [41]; (2) the internal nozzles at the bottom of the digester allow high pressure flow of the biogas, which enhances mixing; and (3) the process operates with a high TS content of 25% to 35% [51]. This system was specifically developed for the treatment of organic wastes; however, the operating parameters have not yet been optimized for treating straw [57].

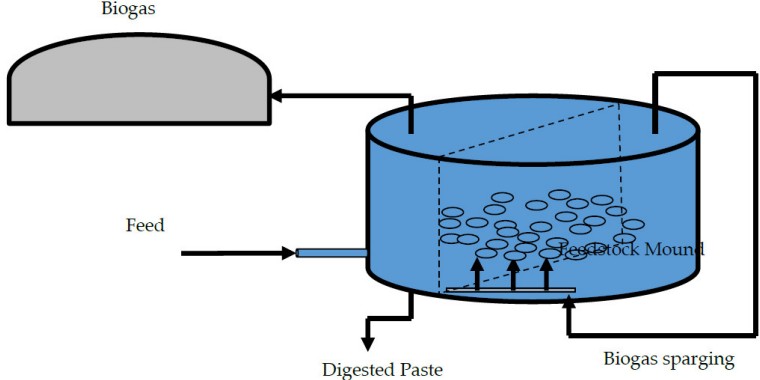

**Figure 8.** Schematic diagram of the continuous Valorga process.

### 2.3.4. Chinese Continuous Reactors

Vertical plug flow (VPD) digestion and two-phase anaerobic digestion (TPAD) are the main continuous processes used in China [54,58]. The VPD system is a step-wise process where the new feedstock is added on top of the previous feedstock, which descends vertically during digestion. VPD reactors contain internal spray systems for recycling the leachate, which is beneficial for inoculating the new substrate with the desired bacteria [59]. The TPAD process is separated into acidogenic and methanogenic phases. The effluent liquid of the hydrolysate flows into the methanogenesis system for biogas production; then, the slurry is pumped back to the acetogenesis system for inoculation [59]. However, straw requires physical pre-treatment to achieve adequate fluidity, where a particle size of < 2.0 cm is usually suitable. This technology is still mainly in the pilot-project stage, and only a few medium-to-large-scale biogas plants exist [25,55].

### 2.3.5. Comparison in Technology

Table 2 compares continuous digestion processes, where the following conclusions were drawn: (1) Mechanical pretreatment (e.g., milling) is required to avoid blockages in the reactor; (2) the feasible OLR of the Valorga system is higher than that of the Linde-KCA, Dranco, and Chinese continuous DADT systems, however, the optimal parameters for treatment of straw are unknown for the Valorga system. Hence, the Linde-KCA system is thought to be more appropriate for treating agricultural straw; (3) the digestion temperature of the Dranco system is higher than that of the others and it requires more energy to maintain stable operation; (4) leachate recirculation or mechanical stirring is necessary to improve mass transfer of the substrate and microorganisms; (5) continuous digestion systems usually have more stable and continuous biogas production than batch system; (6) currently, the Chinese technology is still in the piloted research stage.

**Table 2.** Comparison of the performance of different continuous digestion processes.

| Reactors | Country | Capacity (t/year) | Substrate | TS (%) | OLR (kg VS/(m$^3$·d) | T (°C) | HRT (d) | Methane Yield (Nm$^3_{CH4}$·kg$_{vs\ removed}^{-1}$) | Methane Average (%) | Ref. |
|---|---|---|---|---|---|---|---|---|---|---|
| Dranco | BE | 10,000–70,000 | AW, OFMSW | 10–32 | 10~15 | 50–55 | 20 | 0.21–0.30 | 50 | [32, 50] |
| Valorga | FRA | 20,000–350,000 | AW, OFMSW | 36–60 | 10~15 | 37–55 | 20–33 | 0.21–0.30 | 50–55 | [32, 50] |
| Linde-KCA | GER | N/A | AW, OFMSW | 15–45 | N/A | 37–55 | N/A | N/A | 55 | [38] |
| VPF | PRC | N/A | AW | 8–10 | N/A | 37 | N/A | N/A | 50 | [42] |
| TPAF | PRC | N/A | AW, OFMSW | ≥8 | N/A | 30 | 25 | N/A | N/A | [42] |

N/A: Not available, AW: Agricultural waste, OFMSW: Organic fraction of municipal solid waste, T: Fermentation temperature, HRT: Hydraulic retention time.

## 3. Feasibility in China

In some countries, such as Germany, France, the use of DADTs for producing biogas has been promoted for a long time [60]. China is gradually encouraging the development and implementation of dry digestion projects. Identifying the appropriate DADTs, according to the distribution of crop straw, yield and geographical conditions in China, are used to absorb local straw resources and improve the utilization efficiency of straw. That is also essential for increasing production efficiency and reducing costs to strengthen the competitiveness of commercial operation [61,62]. It should consider the availability of agricultural wastes and the state of biogas demand. DADTs seems to a feasible method to realize the efficient utilization the low value agricultural waste, which can improve environmental and economic sustainability and create a sustainable energy communities network [63].

### 3.1. Collection of Agricultural Residues

The development and implementation of batch and continuous DADTs are limited by resource distribution, the amount of available feedstock, and geographical conditions. Magdalena et al. propound that the factor affecting the achievement of high eco-efficiency is the location of a biogas plant, from which the basic feedstock for biogas production is supplied [64]. In China, most agricultural production is for human food, especially grains (including rice, wheat, and corn). The highest to straw production (24.37%) is from the Mid-South district, where Henan has the largest contribution of 10.37%. This is followed by the East district (23.80%) with the largest contribution coming from Shandong (8.82%). The third district is the North East (18.20%) where the major contributions come from Heilongjiang (8.55%) and Jilin (5.92%) [65]. Continuous DADTs are thought to be suitable for the expending digestion applications in order to fully utilize local straw resources and avoid excessive occupation of farmland, which also can provide biogas for farmers. In the regions able to provide sustainable, stable, and long-term straw resources, continuous digestion technology can be used to produce a constant supply of renewable energy. Batch digestion technology would be preferable in areas with relatively poor straw resources as the batch volume of anaerobic digestion can be adjusted accordingly. Hence, it is easier to manage variability in straw resources due to variations in straw cultivation.

### 3.2. Biogas Utilization

With the aim of developing renewable energy and reducing greenhouse gas emissions, legislation regarding biogas as an alternative energy source needs to be implemented in China [66]. The most common use of biogas in China is in household cooking, where methane is converted to heat energy. When even a large amount of continuous supplies of biogas could be achieved, it is expected that commercial applications would be the main utilization way in future [67]. Therefore, applying DADTs can promote the development of clean energy.

Currently, it is important to explore the utilization of biogas to meet the particular requirements of various usage strategies. First, to guarantee a constant supply of biogas, batch and continuous DADTs should be used for different applications to take advantage of the specific features of each technology.

In the case of batch digestion technology, several digesters can be set up in a plant as needed. The peak and off-peak phases of biogas production of each digester should be overlapped to ensure continuous production. In the future, electricity and heat generation via combined heat and power units is expected to be the main use of biogas, while bio-methane can be injected into the gas grid, and/or converted into compressed or liquefied natural gas for transport fuels [68]. Katharina Bär et al. proved that two-stage high-pressure digestion process can change the methane volume fraction from 75% to 90% by providing pressurized biogas at 0.5 bar [69]. Although it is still in the laboratory stage, it may be a good choice for the commercialization of biogas in future.

## 4. Conclusions and Suggestions

It has been widely accepted that providing sufficient, sustainable, and affordable energy to China is particularly important for mitigating further environment problems. This paper has introduced some critical indicators and challenges regarding DADTs and their suitability in China. Batch reactors have the advantage of being a relatively simple and robust technology with low maintenance requirements. However, the stability and continuity of the biogas supply can be more easily guaranteed using continuous reactors, in spite of higher maintenance and management requirements. Batch anaerobic digestion processes would be more effective in southern China due to the prevalence of small-scale agriculture, while continuous anaerobic digestion would be preferable in the north where large-scale agriculture is common. To improve the flexibility, adaptability, and efficiency of DADTs, further research and technological improvements are required.

The DADTs, taking into account the features of the analyzed urban area, play a key role in determining the waste-to-energy opportunity. Meanwhile, the supply and characteristics of the agricultural straw available should be considered before, specific technology is determined. This article contributes to the discussion on the sustainable development of biogas, high-value utilization of agricultural waste and economic development of rural energy, and hopes to provide a new mentality for high-value utilization of agricultural waste in other countries.

**Author Contributions:** Y.F. assisted in dry anaerobic digestion system analysis and wrote the manuscript. T.L. performed analysis of every system. Z.M. and J.L. designed the study and assisted in data analysis. K.Q. and Y.G. assisted in manuscript editing and article review.

**Funding:** This research was funded by the National Key Technology Support Program (2015BAD21B03), the National Natural Science Foundation of China (201802180094), Agricultural Science and Technology Innovation Program of the Chinese Academy of Agricultural Sciences, the Sichuan Center for Rural Development Research (C R1610).

**Acknowledgments:** The authors gratefully acknowledge financial support from the National Key Technology Support Program (2015BAD21B03), the National Natural Science Foundation of China (201802180094), Agricultural Science and Technology Innovation Program of the Chinese Academy of Agricultural Sciences, the Sichuan Center for Rural Development Research (C R1610).

**Conflicts of Interest:** The authors declare no conflict of interest.

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
