# Peer review of "Dry Anaerobic Digestion Technologies for Agricultural Straw and Acceptability in China"

_sustainability, doi:10.3390/su10124588_

Round 1
Reviewer 1 Report
In the present manuscripts, the authors present a review about utilizing agricultural straw as feedstock for biogas production. The overall content of the manuscript is OK. But many small points should be revised carefully before further processing. See below.
Line 34, why does biogas seem to be the most effective way to treat organic waste? In addition to biogas, Many other renewable ways to utilize the agricultural wastes (biobutanol, or bioethanol) refer to the following publications. You may want to highlight the advantage of biogas over biobutanol or bioethanol.
https://www.sciencedirect.com/science/article/pii/S1369703X18300986?via%3Dihub
Line 41, wording, “is”?
Line 44-54, Suggest to add reviews about the advantage of dry fermentation.
Line 68, “make it is very difficult to digest very well.” Wording “is”
Author Response
Responses to Reviewer 1
Responses to specific comments
Points 1: Line 34, why does biogas seem to be the most effective way to treat organic waste? In addition to biogas, Many other renewable ways to utilize the agricultural wastes (biobutanol, or bioethanol) refer to the following publications. You may want to highlight the advantage of biogas over biobutanol or bioethanol.
Response 1: Thank you very much for your suggestions. Based on your comment, we have increased other renewable ways for agricultural wastes utilization (see lines 34-37 in revised version). We describe the advantage of biogas over biobutanol or bioethanol. As biogas seems to be a more effective way to produce energy in China, due to low operation level, compared with other renewable ways, otherwise, biogas technology could be a practically feasible process at many scales, from a household to large-scale biogas plants. in actually, the energy yield of bioethanol from agricultural waste is less than that of biogas in most circumstance .
Points 2: Line 41, wording, “is”?
Response 2: Thank you very much for your suggestions. Correction has been done (see line 43-44 in revised version).
Points 3: Line 44-54, Suggest to add reviews about the advantage of dry fermentation.
Response 3: Thank you very much for your comment. the advantage of dry anaerobic digestion has been supplemented. Four bottlenecks could be overcame directly, including extra water requirement, larger volume, low TS content floating. What’ more, some technologies won’t require the reduction of size. (see lines 55-57 in revised version)
Points 4: Line 68, “make it is very difficult to digest very well.” Wording “is”
Response 3: Thank you very much for your suggestion. However, it has been revised according. (see line 79-80 in revised version).
Reviewer 2 Report
As the authors highlighted in this article, finding a sustainable solution for converting underutilized agricultural residues to useful and renewable energy is critical for China. Therefore a review article in this field will be a great contribution. However, the article needs an extensive revision due to poor writing quality in both language and style. It needs an overall editing (grammar, structure and style-wise).
The structure and the logic of the article were not easy to follow and understand, it needs to be revised.
Author Response
Responses to specific comments
Points 1: However, the article needs an extensive revision due to poor writing quality in both language and style. It needs an overall editing (grammar, structure and style-wise).The structure and the logic of the article were not easy to follow and understand, it needs to be revised.
Response: Thank you very much for your useful suggestion, the linguistic problem has been polished.
(1) Based on your comment, an overall more careful editing has been done by language editing company and the authors.
(2) In the revised version, the logical between different parts of article is added (see in section 1 or lines 61-68). In section 2, we increase explanation between the indicators and DADTs. Reasons are written as “DADTs must meet the optimal straw digestion parameters, which can affect the cost of project operation and the efficiency of gas production” (see in section 2 or lines 73-75). In section 3, we add the description of the necessary to popularize DADTs in China. DADTs could be used to accomplish the high-value utilization of agricultural waste, and then put forward to the acceptability of DADTs in China (see lines 398-400 and lines 401-44). In section 4, the conclusion has been modified. The conclusion is revised as “The DADTs, taking into account the features of the analyzed urban area, plays a key role in determining the waste-to-energy opportunity. Meanwhile, the supply and characteristics of the agricultural straw available should be considerable when determine the choice of a specific technology. This article contributes to the discussion on the sustainable development of biogas, high-value utilization of agricultural waste and economic development of rural energy and hope to provide a new mentality for high-value utilization of agricultural waste in other countries.”
Reviewer 3 Report
Dear Authors,
attached you can find the manuscipt with some revision.
I suggest a strong revision of the English in order to improve legibility.

Author Response
Responses to Reviewer 3
Responses to specific comments
Points 1: I suggest that a strong revision of the English in order to improve legibility.
Response 1: Based on your comment, an overall more careful editing has been done by language editing company and the authors.
Points 2: Why fermentation? It’s anaerobic digestion, I think that probably it is better to prefer the review to “dry anaerobic digestion”.
Response 2: Thank you very much for your suggestion. The corresponding revision has been done.
Points 3: Line 28: Change “environment” to “environmental”.
Response 3: Thank you very much for your comment. The corresponding revision has been done (see 29 in revised version).
Points 4: Line 41: delete “is” after “fermentation”.
Response 4: Thank you very much for your comment. The related sentence has been modified (see 44 in revised version).
Points 5: Line 75: “Pretreatment” isn’t an indicator. Probably the degree of hydrolysis could be used as an indicator.
Response 5: Thank you very much for your suggestion “The extent of hydrolyzation)” is used to replace “Pretreatment”. Pretreatment methods are used to accelerate the degradation of raw materials. So we continue to describe pretreatment methods in this section (see lines 87-90 in revised version).
Points 6: Line 256: Change “HTR” to “HRT”.
Response 6: Thank you very much for your comment. The corresponding revision has been done (see lines 283 in revised version).
Points 7: Line 256: Could you add some data biogas production?
Response 7: Thank you very much for your comment. We add the “Methane yield” and “Methane average” in Table 1 and Table 2 to describe the biogas-producing performances.
Points 8: Line 340: The inner wall is not well descripted in the figure: the reactors contain a vertical median inner-wall on approximately 2/3 of their diameter.
Response 8: Thank you very much for your suggestion. In revised version, more detail description has been added (see lines 342-343)
Reviewer 4 Report
The paper is interesting and cover great part of the available literature.
Please avoid lumping references such as in lines 31 and 37.
To improve especially the estimations in Tables 1 and 2 as well as the logic behind the need for going further with biomass and bioenergy, consider the following studies in this Journal and other high-ranked ones:
https://www.mdpi.com/2071-1050/10/10/3720
https://www.mdpi.com/1996-1073/10/2/229
https://www.mdpi.com/2076-3417/8/11/2083
https://www.sciencedirect.com/science/article/pii/S0961953411005563
https://www.sciencedirect.com/science/article/pii/S0961953418300989
https://www.sciencedirect.com/science/article/pii/S0960148115002396
What is the research gap to fill and the research question formulated by the authors?
What are the outcomes imported from other Countries and what result can be exported to other Countries from your study?
Please enrich the conclusions by summarizing the research question, data and methods used, the results with perspectives and limitations.
Author Response
Responses to Reviewer 4
Responses to specific comments
Points 1: Please avoid lumping references such as in lines 31 and 37.
Response 1: Thank you very much for your suggestion. The relevant references have been modified thoroughly.
Points 2: To improve especially the estimations in Tables 1 and 2 as well as the logic behind the need for going further with biomass and bioenergy, consider the following studies in this Journal and other high-ranked ones:
Response 2: Thank you very much for your comment.
We add the “Methane yield” and “Methane content or average methane content” in Table 1 and Table 2 to describe the biogas-producing performances. The basic data of technologies are described for comparison in Table 1 and Table 2. The advantages of biogas over other renewable energy have been illustrated (e.g. biobutanol, or bioethanol) (see lines 34-47 in revised version). In section 3, creating a sustainable energy communities network is introduced to improve environmental and economic sustainability (de Santoli, Mancini et al. 2015), it is the purpose to extend DADFs (see lines 301-404 in revised version). The location of a biogas plant is the key factor to accomplish high eco-efficiency (Muradin, Joachimiak-Lechman et al. 2018), so location and scale of DADFs are closed to the distribution and amount of agricultural waste (see lines 407-409 in revised version). For biogas utilization, the original version mention some maners to use biogas, besides that the study of Bär, Merkle et al. (2018) has been added to this paper (see lines 437-438 in revised version).
Points 3: What is the research gap to fill and the research question formulated by the authors?
Response 3: Thank you very much for your suggestion. DADTs have not been broadly popularized yet, and agricultural waste is not used effectively. The purpose of this paper is used for promoting the acceptability of DADTs in China, which provide a maneuverable and economical method for agricultural waste utilization. The purpose and main contents of each section are described in more detail, in order to more conveniently understand the structure (see line 61-78 in revised version).
Points 4: What are the outcomes imported from other Countries and what result can be exported to other Countries from your study?
Response 4: Thank you very much for your suggestion. We hope this development model is developed in China in advance, and then provide a new mentality for high-value utilization of agricultural waste in other countries (see lines 455-457 in revised version). The related sentences are modified as “This article contributes to the discussion on the sustainable development of biogas, high-value utilization of agricultural waste and economic development of rural energy and hope to provide a new mentality for high-value utilization of agricultural waste in other countries. ”
Points 5: Please enrich the conclusions by summarizing the research question, data and methods used, the results with perspectives and limitations.
Response 5: Thank you very much for your suggestion. The conclusion has been modified thoroughly. Please see section 4 in revised version.
Round 2
Reviewer 1 Report
After revision, the quality of the manuscript has been improved to be a solid publication.
Author Response
Thank you very much for your suggestion